# Abundant, diverse, unknown: Extreme species richness and turnover despite drastic undersampling in two closely placed tropical Malaise traps

**Caroline Chimeno**[1] *, **Stefan Schmidt**[1], **Bruno Cancian de Araujo**[1,2], **Kate Perez**[3], **Thomas von Rintelen**[4], **Olga Schmidt**[1], **Hasmiandy Hamid**[5], **Raden Pramesa Narakusumo**[6], **Michael Balke**[1]

1 Zoologische Staatssammlung München (SNSB-ZSM), Munich, Germany, 2 Entomological Biodiversity Laboratory, Federal University of Espirito Santo, Vitoria, Brazil, 3 Centre for Biodiversity Genomics, University of Guelph, Guelph, Ontario, Canada, 4 Center for Integrative Biodiversity Discovery, Museum für Naturkunde—Leibniz-Institut fur Evolutions- und Biodiversitätsforschung, Berlin, Germany, 5 Department of Plant Protection, Faculty of Agriculture, Universitas Andalas, Padang, Indonesia, 6 Research Center for Biosystematics and Evolution, National Research and Innovation Agency (BRIN), Museum Zoologicum Bogoriense, Cibinong, Indonesia

* chimeno@snsb.de

**Data Availability Statement:** All original data spreadsheets that were downloaded from BOLD for analysis have been uploaded to Figshare (https://

## Abstract

Arthropods account for a large proportion of animal biomass and diversity in terrestrial systems, making them crucial organisms in our environments. However, still too little is known about the highly abundant and megadiverse groups that often make up the bulk of collected samples, especially in the tropics. With molecular identification techniques ever more evolving, analysis of arthropod communities has accelerated. In our study, which was conducted within the Global Malaise trap Program (GMP) framework, we operated two closely placed Malaise traps in Padang, Sumatra, for three months. We analyzed the samples by DNA barcoding and sequenced a total of more than 70,000 insect specimens. For sequence clustering, we applied three different delimitation techniques, namely RESL, ASAP, and SpeciesIdentifier, which gave similar results. Despite our (very) limited sampling in time and space, our efforts recovered more than 10,000 BINs, of which the majority are associated with "dark taxa". Further analysis indicates a drastic undersampling of both sampling sites, meaning that the true arthropod diversity at our sampling sites is even higher. Regardless of the close proximity of both Malaise traps (< 360 m), we discovered significantly distinct communities.

## Introduction

In the age of rapid biodiversity decline, taxonomists find themselves in a race against time to discover and describe new species before they become extinct [1–5]. However, identifying species in several megadiverse groups of organisms requires in-depth taxonomic expertise, which

doi.org/10.6084/m9.figshare.21815034). The R
script and all input data sets are also deposited on
Figshare (R code: https://doi.org/10.6084/m9.
figshare.21806370.v2; BIN dataset: https://doi.org/
10.6084/m9.figshare.21815142; ASAP dataset:
https://doi.org/10.6084/m9.figshare.21815064.v1).
The datasets on BOLD can be found under doi.org/
10.5883/DS-GMTINDO1 and doi.org/10.5883/DS-
GMTINDO2.

**Funding:** The project was supported by the
Deutsche Forschungsgemeinschaft and the
Bundesministerium für Bildung und Forschung
(BMBF) within the bilateral "Biodiversity and Health"
funding program (Project numbers: 16GW0111K,
16GW0112) with additional support from DIPA
PUSLIT Biologi LIPI 2015-2016. The sequence
analyses for this study were supported, in part, by
Genome Canada through the Ontario Genomics
Institute, while informatics support was provided
through a grant from the Ontario Ministry of
Research and Innovation. The funders did not play
any role in the study design, data collection and
analysis, decision to publish, or preparation of the
manuscript.

**Competing interests:** The authors have declared
that no competing interests exist.

is either in decline or very limited, the latter being the case in the so-called dark taxa [6, 7]. This mismatch between high species numbers awaiting discovery and few researchers available for doing so is also known as the "taxonomic impediment". It is prominent among arthropods [8], and considering that arthropods account for a large proportion of the animal biomass and diversity in terrestrial ecosystems [9–11], is a direct constraint to global biodiversity research. Often, ecological surveys must limit their analyses to a subset of known species (e.g., flagship indicator species) because there is not enough know-how to analyze the highly abundant, often minute specimens that make up the bulk of the sample [12, 13].

As a potential remedy, molecular identification techniques have greatly evolved in the last decade, providing accelerated sample processing methodologies in various fields of research [14]. DNA barcoding, for example, is a method that uses a short DNA sequence of the COI gene in the mitochondrial DNA to identify and distinguish species from one another [15–17]. Paul Hebert and colleagues first introduced it in 2003, and today, it is a standard approach for molecular identification or presorting species [18]. DNA barcoding is easy to use [even for non-experts], widely available, and nowadays economic [16, 19–22].

In 2012, the Global Malaise trap Program was initiated by the Centre for Biodiversity Genomics (CBG) at the Biodiversity Institute of Ontario (BIO) with the large-scale worldwide deployment of Malaise traps (see https://biodiversitygenomics.net/site/projects/gmp/). Malaise traps are very efficient at capturing flying insects and are, therefore, commonly used in surveys of terrestrial arthropods [23–26]. More than 158 sites in 33 countries were sampled and analyzed via DNA barcoding to provide an overview of the global arthropod biodiversity and provide detailed temporal and spatial information on arthropod communities (see https://biodiversitygenomics.net/site/projects/gmp/). In a joint project with the Andalas University, two Malaise traps were deployed in Padang, Sumatra, Indonesia, and operated for three months each. Insect communities in tropical regions are notorious for being extraordinarily diverse [11, 27, 28] yet severely understudied [29, 30], making the large-scale sequencing of the Malaise traps contents especially interesting. In this study, we present and evaluate the sequencing results recovered for each Malaise trap.

## Materials and methods

### Collecting

In 2016, we deployed two Malaise traps, installed ca. 360 m apart from each other. We set up the traps at the northern forest edge of the 500-hectare campus area of the University of Andalas at the eastern part of Padang City, West Sumatra Province, Indonesia (Fig 1). The traps were located in a semi-open area dominated by ferns, interspersed with medium-sized and a few large-sized trees. The trap locations were set up in spots with sparse vegetation in such a way that flight paths were open in both directions of the traps. The adjacent tropical forest was dominated by secondary tree vegetation and is connected to the Bukit Barisan mountain range. Both Malaise traps were operated from May 5th to July 30th. The collection bottles were emptied biweekly and topped up with fresh 80% EtOH. Samples were stored in a freezer until further processing. Because both traps were located on the grounds of the university, no collection permit was needed.

### Sample processing

All collection bottles were sent to the Centre for Biodiversity Genomics for sorting and further processing. An attempt was made to barcode as many specimens as possible, but due to funding constraints, not all specimens were processed, and for some collection bottles, a maximum of fifteen 96-well microplates were filled (Table 1). A selective strategy was used

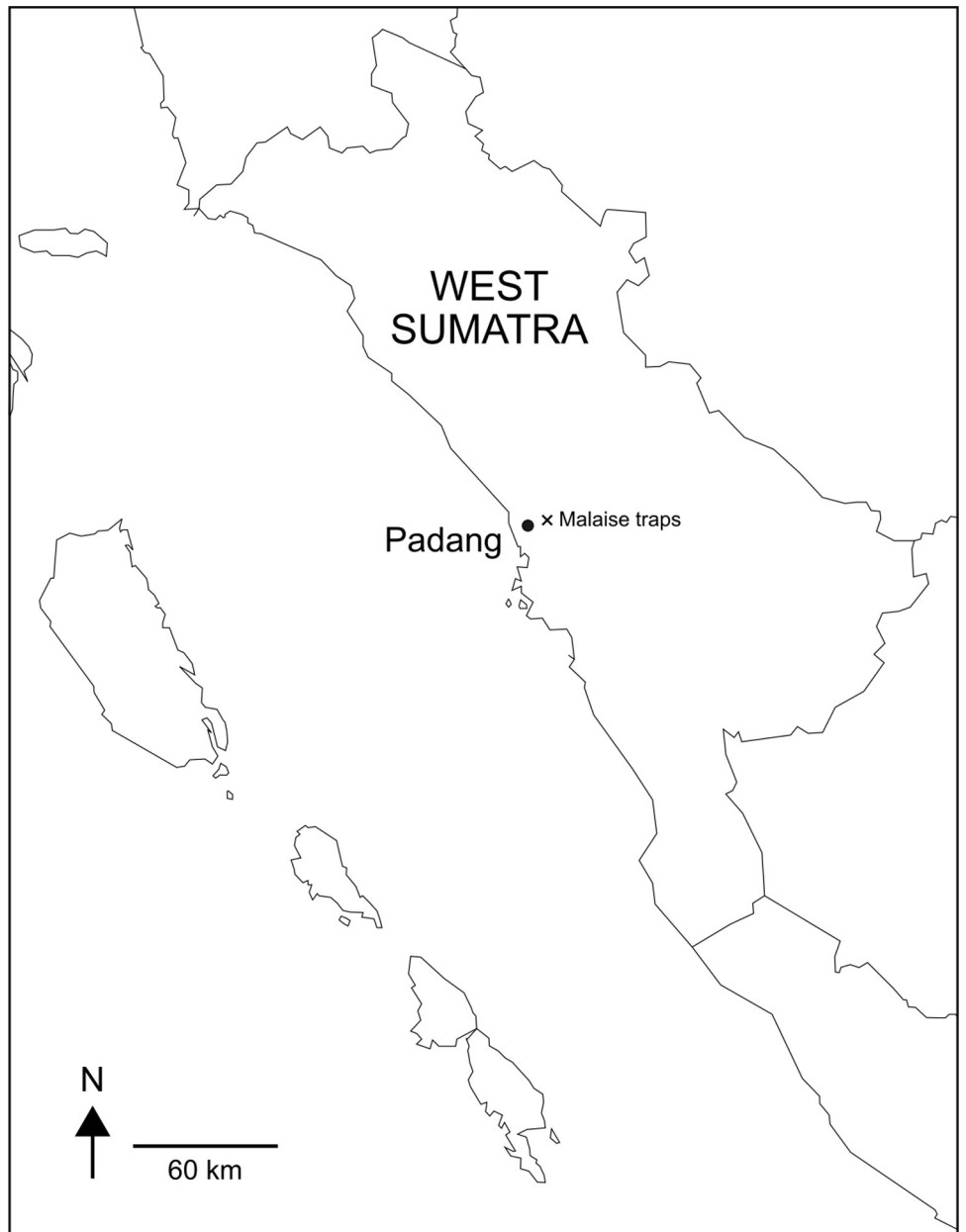

**Fig 1. Collection sites.** Malaise trap sites near Padang, West Sumatra. Created by the authors using QGis.

to narrow down the number of specimens especially for the samples with only 15 plates processed. Individuals chosen for sequencing were selected to capture as much diversity as possible based on size and morphospecies. Two sizes of sieves were used to subsample from three different size classes (no sieve, 8mm sieve, and 2mm sieve). As most of the diversity was likely hidden in the smaller organisms (particularly the abundant insect orders: Hymenoptera, Coleoptera, and Diptera), more specimens were chosen from the smallest size class. Additionally, more Hymenoptera and Coleoptera were selected as opposed to Diptera because Diptera are often so abundant in Malaise trap samples that there is a higher risk of oversampling the same species.

**Table 1. Malaise trap sample information.** Collection dates, sequencing capacity, specimens processed, and sequences obtained per sample.

| Malaise Trap | Sample Nr. | Collection Interval | Sequencing capacity | Nr. Specimens | Nr. Sequences |
|---|---|---|---|---|---|
| Trap 1 | 1 | 05–13 May | All specimens | 5,969 | 5,101 |
| Trap 1 | 2 | 13–20 May | 15 plates | 1,475 | 1,232 |
| Trap 1 | 3 | 20–27 May | All specimens | 9,070 | 5,836 |
| Trap 1 | 4 | 27 May–03 June | 15 plates | 1,475 | 1,082 |
| Trap 1 | 5 | 03–10 June | All specimens | 9,886 | 7,760 |
| Trap 1 | 6 | 10–24 June | 15 plates | 1,475 | 1,325 |
| Trap 1 | 7 | 24 June–01 July | All specimens | 6,266 | 5,730 |
| Trap 1 | 8 | 01–08 July | 15 plates | 1,475 | 1,216 |
| Trap 1 | 9 | 08–15 July | All specimens | 4,230 | 3,776 |
| Trap 1 | 10 | 15–22 July | 15 plates | 1,475 | 1,267 |
| Trap 1 | 11 | 22–30 July | All specimens | 9,566 | 8,567 |
| Trap 2 | 1 | 05–13 May | All specimens | 10,102 | 8,439 |
| Trap 2 | 2 | 13–20 May | 15 plates | 1,475 | 1,229 |
| Trap 2 | 3 | 20–27 May | 15 plates | 1,504 | 1,136 |
| Trap 2 | 4 | 27 May–03 June | 15 plates | 1,475 | 1,169 |
| Trap 2 | 5 | 03–10 June | 15 plates | 1,529 | 1,250 |
| Trap 2 | 6 | 10–24 June | 15 plates | 1,475 | 1,349 |
| Trap 2 | 7 | 24 June–01 July | 15 plates | 1,490 | 1,241 |
| Trap 2 | 8 | 01–08 July | 15 plates | 1,475 | 1,280 |
| Trap 2 | 9 | 08–15 July | 15 plates | 1,491 | 1,285 |
| Trap 2 | 10 | 15–22 July | 15 plates | 1,475 | 1,272 |
| Trap 2 | 11 | 22–30 July | 15 plates | 1,644 | 1,258 |

Tissue lysis was performed overnight at 56°C, and DNA extraction was conducted using an automated, silica membrane-based protocol [31]. To reduce costs and the amount of reagents needed for PCR amplification of the COI gene, the DNA extracts from four 96-well plates were consolidated into 386-well PCR plates [23, 32]. The PCR products were diluted, unidirectionally sequenced, then cleaned-up using an automated magnetic bead-based method before being sequences on an ABI 3730 xl DNA Analyzer (Applied Biosystems). For more details on the laboratory protocols, see [23].

All barcoded specimens are currently stored at the Center for Biodiversity Genomics (CBG) natural history archive (collection code BIOUG) at the University of Guelph, Canada. However, this collection, as well as the rest of the unprocessed material, will eventually be repatriated to Museum Zoological Bogoriense in Cibinong, Indonesia.

### Data analysis

All specimen metadata and sequence data were uploaded to the Barcode of Life Data System (BOLD), an online workbench and database [32]. All data is publicly available on BOLD in two datasets (doi.org/10.5883/DS-GMTINDO1 and doi.org/10.5883/DS-GMTINDO2). We also uploaded the BOLD data spreadsheet including all metadata of specimens to Figshare.

Sequences were assigned a Barcode Index Number (BIN) by the BOLD system using the RESL-algorithm. BINs represent globally unique identifiers for clusters of sequences as a species proxy [32]. Every sequence $\geq$ 300 base pairs (bp) is automatically assigned to a Barcode Index Number (BIN) that is already in BOLD if sequence similarity based on the RESL-algorithm is fulfilled [32]. Sequences with $\geq$ 500 bp which do not find a match, serve as founders

of new BINs. Family-level identifications were conducted using the BIN taxonomy match tool on BOLD.

All analyses were performed in R version 4.2.1 [33], using the packages *vegan* version 2.5–7 [34], *iNEXT* version 2.0.20 [35], and *SpadeR* version 0.1.1 [36]. To assess our sampling effort, we created accumulation curves of BINs for each Malaise trap (via *iNEXT; iNEXT* package) and estimated the species diversity present at each sampling site (via *ChaoSpecies; SpadeR* package). We created continuous diversity profiles for each trap (via *Diversity; SpadeR* package) to illustrate the variation in the three standard metrics of biodiversity that are quantified by Hill numbers (q): species richness (q = 0), Shannon diversity (q = 1), and Simpson diversity (q = 2). Hill numbers are a mathematically consolidated group of diversity indices that include relative species abundances to quantify biodiversity. To evaluate the faunal similarity between Malaise traps, we performed permutation multivariate analysis of variance (PERMANOVA) (via *adonis2*; *vegan* package; Bray Curtis dissimilarity; 999 permutations). We differentiated between location and dispersion effects by applying a beta dispersion test analogous to Levene's test (via *betadisper*; *vegan* package) and an *F*-test (via *permutest*; *vegan* package). For visualization, we created a non-metric dimensional scaling (NMDS) ordination (via *metamds*; *vegan* package; Bray Curtis dissimilarity). Using the universal insect trait tool (ITT; version 1.0) [37], we categorized all arthropod families into ecological guilds to analyze differences of the functional diversity between the insect communities of the two trap sites in addition to their taxonomic diversity.

Because the BIN concept has been challenged recently [38], we decided to compare the number of OTUs recovered with other clustering algorithms. BINs should not be considered synonymously of "species", but rather as a dynamic tool to presort the global DNA barcode database into MOTUs that taxonomists can further evaluate; BIN definitions might change on BOLD as more sequences are added to the database. Since the assignment of BINs in a dataset is affected by other sequences in BOLD that are not included in the dataset, we analyzed the sequences of our datasets using the "Cluster Sequences" option in BOLD. This way, the resulting OTUs are directly comparable to the results of other species delimitation algorithms. As a consequence, the number BINs found in our project on BOLD are slightly higher than in our analyses because the system assigns BINs to sequences between 300 and 499 bp if the BIN is already present in the database, whereas we limited analyses to sequences displaying a minimum length of 500 bp. In addition to RESL, we analyzed our data using the Assemble Species by Automatic Partitioning program (ASAP) [39] using the web interface, and we analyzed the same data using SpeciesIdentifier version 1.9 [40]. ASAP employs pairwise genetic distances for hierarchical clustering without using the information on intraspecific diversity, and SpeciesIdentifier is an algorithm that allows clustering sequences based on their pairwise genetic distances (p-distances). To visualize the outputs of the different clustering algorithms (RESL, ASAP, SpeciesIdentifier), we created accumulation curves (via *iNEXT; iNEXT* package) depicting the number of clusters obtained for each Malaise trap. Detailed specimen and sequence data are accessible in BOLD as two citable datasets (doi.org/10.5883/DS-GMTINDO1 and doi.org/10.5883/DS-GMTINDO2).

## Results

### Alpha-diversity assessments

We obtained 39,374 COI-sequences from Malaise trap 1, and 19,394 for Malaise trap 2 which led to the recovery of 6,177 and 5,206 BINs respectively. Together, we obtained a total of 9,212 BINs, with 2,171 being shared between traps. More than two-thirds (6,125) of all BINs were unique to BOLD, meaning that they were added for the first time with the upload of these

sequences. Of the 58,769 specimens that were successfully sequenced, only 961 automatically obtained a species-level identification, providing coverage for 231 species. The majority of sequences provided identification only to the family level (94%), and most of these were associated to families of insects that are reknown for being challenging to study and therefore highly underrepresented in databases (see Discussion). In this study, eight families of dark taxa were largely represented among our data, namely Cecidomyiidae (gall midges), Ceratopogonidae (biting midges), Chironomidae (non-biting midges), Phoridae (scuttle flies), Psychodidae (sand flies), Sciaridae (dark-winged fungus gnats), Platygastridae and Braconidae (parasitoid wasps). These eight families make up 70% of all specimen numbers, and 58% of all BINs. Fig 2, which presents the frequency of rare and common BINs among the merged dataset, shows that the majority of BINs (66%; 6,078 BINs) were represented by one or two specimens only.

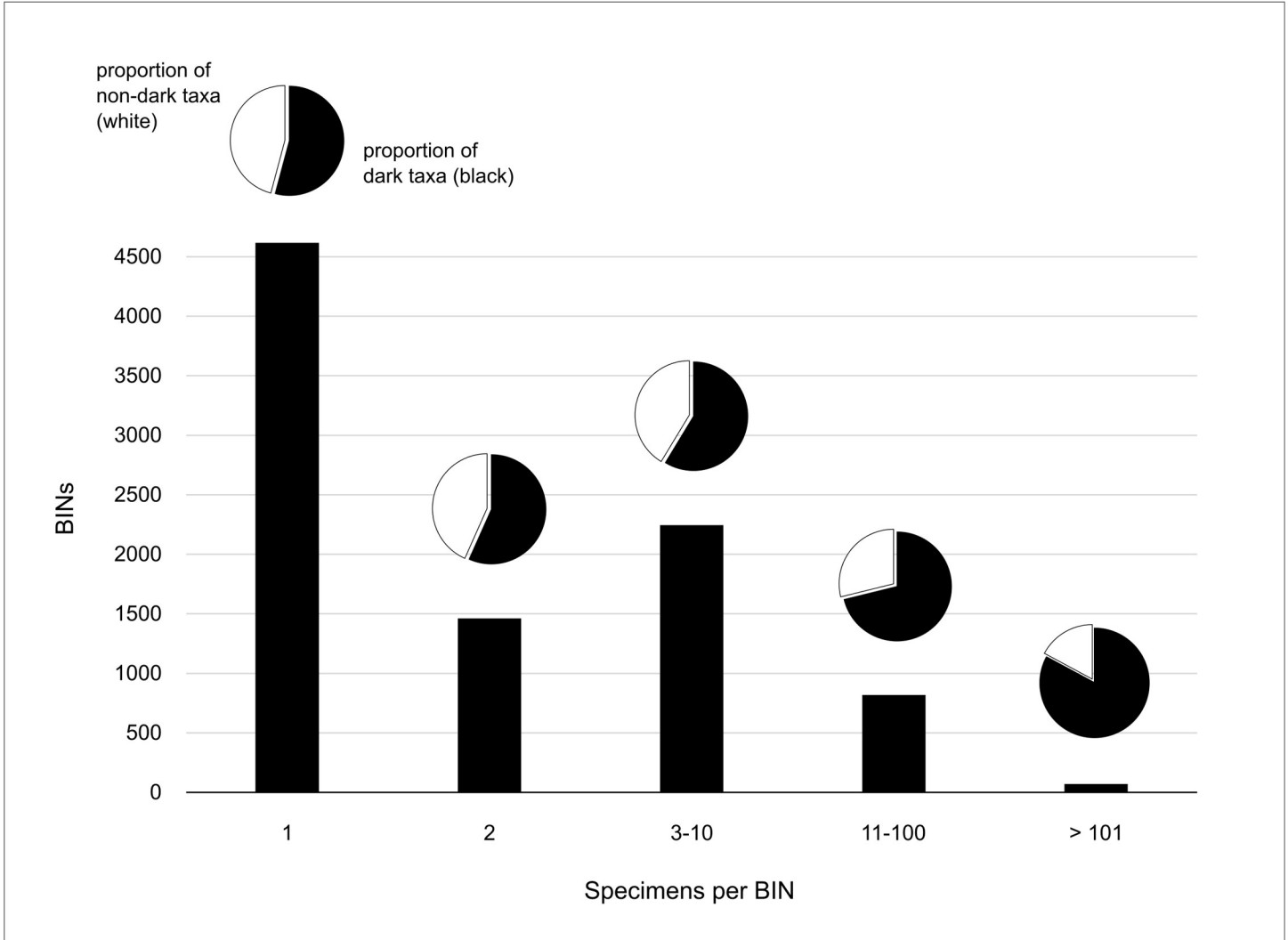

**Fig 2. Frequency of BINs and proportion of dark taxa.** The majority of BINs are rare and are represented by one (BIN frequency = 1) or two (BIN frequency = 2) specimens only. The pie charts represent the proportion of dark taxa among the BIN diversity (in black). These include members of Cecidomyiidae, Ceratopogonidae, Chironomidae, Phoridae, Psychodidae, Braconidae, and Platygastridae.

**Malaise trap 1.** The BINs recovered from Malaise trap 1 provide coverage for 231 families in 21 arthropod orders. The top ten most diverse (from most to least diverse) families are Cecidomyiidae (Diptera; 1,858 BINs), Chironomidae (Diptera; 491 BINs), Ceratopogonidae (Diptera; 470 BINs); Phoridae (Diptera; 439 BINs), Platygastridae (Hymenoptera; 284 BINs); Sciaridae (Diptera; 239 BINs), Psychodidae (Diptera; 145 BINs), Formicidae (Hymenoptera; 125 BINs), Cicadellidae (Hemiptera; 111 BINs), and Braconidae (Hymenoptera; 105 BINs). In total, these families represent 70% of all recovered BINs for this Malaise trap. Chao1 analysis estimated that about 11,000 species may occur at this sampling site, and extrapolation to double the number of captured and processed specimens would have increased the number of recovered BINs to 8,531, which is an increase of 38% (Table 2). In the diversity profile, there is no overlap between the species richness obtained through the analysis of specimens and that estimated to occur at the trap sites (Hill number $q = 0$, Fig 3B).

**Malaise trap 2.** Although we processed substantially fewer specimens from Malaise trap 2, we obtained almost as many BINs (Table 2 and Fig 3A). The BINs from Malaise trap 2 provide coverage for 254 families in 24 arthropod orders. The ten most diverse families are (from most to least diverse): Cecidomyiidae (Diptera; 1,003 BINs), Phoridae (Diptera; 484 BINs), Platygastridae (Hymenoptera; 305 BINs), Sciaridae (Diptera; 220 BINs), Ceratopogonidae (Diptera; 189 BINs), Chironomidae (Diptera; 186 BINs), Cicadellidae (Hemiptera; 158 BINs), Braconidae (Hymenoptera; 152), Erebidae (Lepidoptera; 128 BINs), and Psychodidae (Diptera; 128 BINs). In total, these families represent 86% of all recovered BINs. Chao1 analysis revealed that about 10,000 species might occur at this trap site. Doubling the number of captured specimens would have increased the obtained BIN diversity to 7,481, an increase of 44% (Table 2). As for Malaise trap 1, there is no overlap between the number of empirical BINs obtained from our analyses and the species richness estimated to be present at the site (Fig 3C)

## Beta-diversity analysis

Analysis revealed that 2,171 BINs are shared between both traps, and Chao1-shared estimates suggest that up to 4,281 (± 183) BINs are shared between both communities at the trap sites. PERMANOVA analysis of the sample contents uncovered that the arthropod communities from the Malaise traps are significantly distinct from one another (adonis2 $p = 0.001$) and that

**Table 2. The number of clusters obtained from the COI sequence data of each Malaise trap when applying different clustering algorithms (RESL; ASAP; SpeciesIdentifier).**

| Algorithm | Output | Malaise trap 1 | Malaise trap 2 |
|---|---|---|---|
| RESL OTUs | Number of clusters (n) | 6,283 | 5,253 |
| | Sample coverage | 0.918 | 0.843 |
| | Extrapolation to 2n | 8,699 ± 255 | 7,569 ± 266 |
| ASAP OTUs | Number of clusters (n) | 5,185 | 4,594 |
| | Sample coverage | 0.934 | 0.869 |
| | Extrapolation to 2n | 7106 ± 211 | 6501 ± 213 |
| SpeciesIdentifier OTUs | Number of clusters (n) | 5,967 | 5,054 |
| | Sample coverage | 0.921 | 0.851 |
| | Extrapolation to 2n | 8347 ± 274 | 7260 ± 252 |
| BINs | Number of clusters (n) | 6,177 | 5,206 |
| | Number rare clusters | 4,132 | 4,730 |
| | Chao1 community estimator | 11,280 ± 255 | 10,382 ± 268 |
| | Sample coverage | 0.919 | 0.844 |
| | Extrapolation to 2n | 8,531 | 7,481 |

## a. BIN diversity per Malaise trap

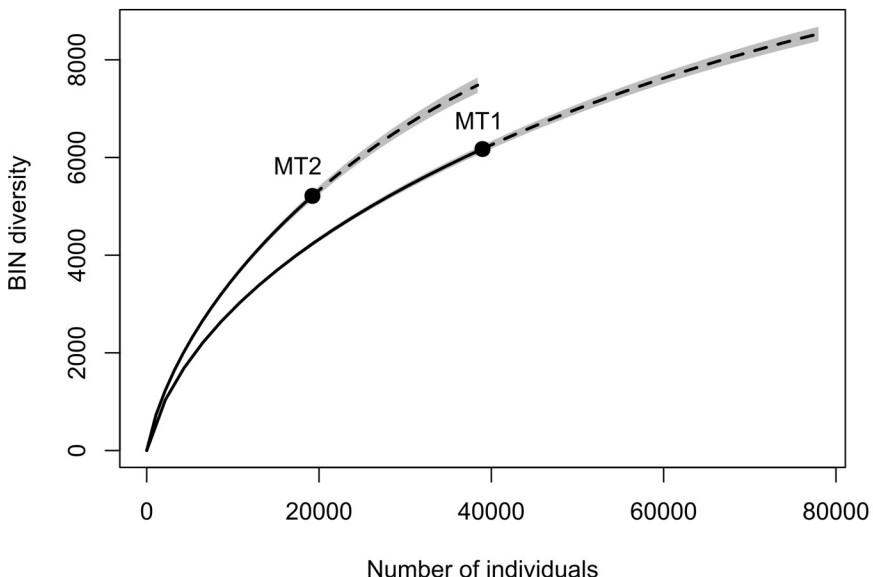

## b. Diversity Profile MT1    c. Diversity Profile MT2

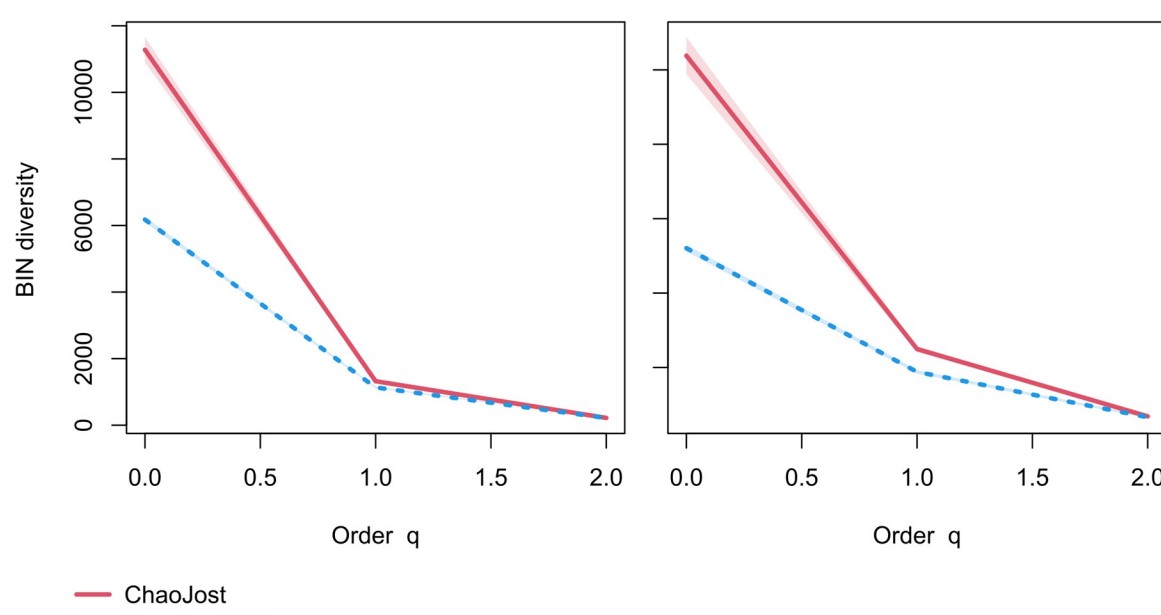

**Fig 3. Accumulation curves of the BIN diversity recovered for each Malaise trap.** Dotted lines represent extrapolated values (up to double the sampling effort), bold lines represent interpolated values. Shaded areas represent the 95% confidence intervals.

this significance is driven by location effects only (S1 Table). In the NMDS ordination, collection samples are clearly clustered based on Malaise trap (S1 Fig). Evaluating the data in more detail, we see that despite high species turnover, both traps depict similar compositions at the family level, which in turn has the same effect on the guild composition (Fig 4A and 4B).

## a. Relative BIN diversity of abundant families

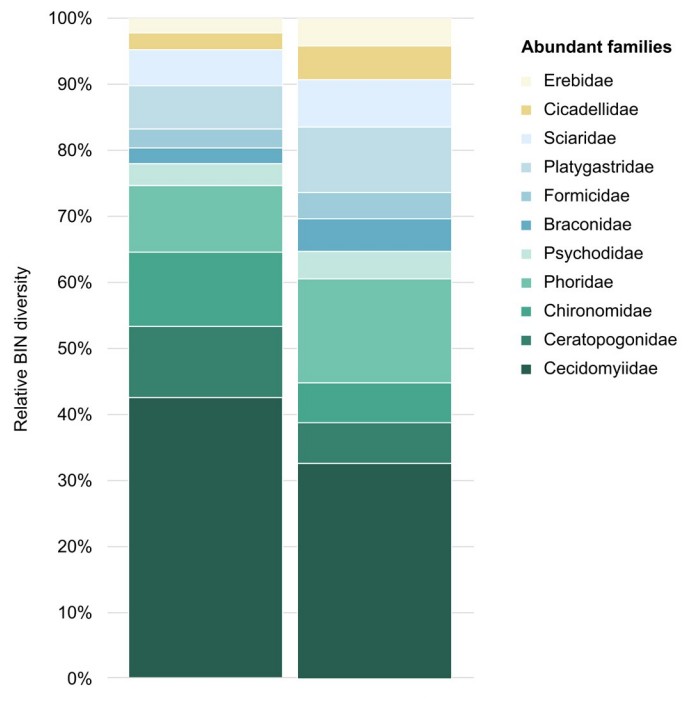

## b. Relative BIN diversity per ecological guild

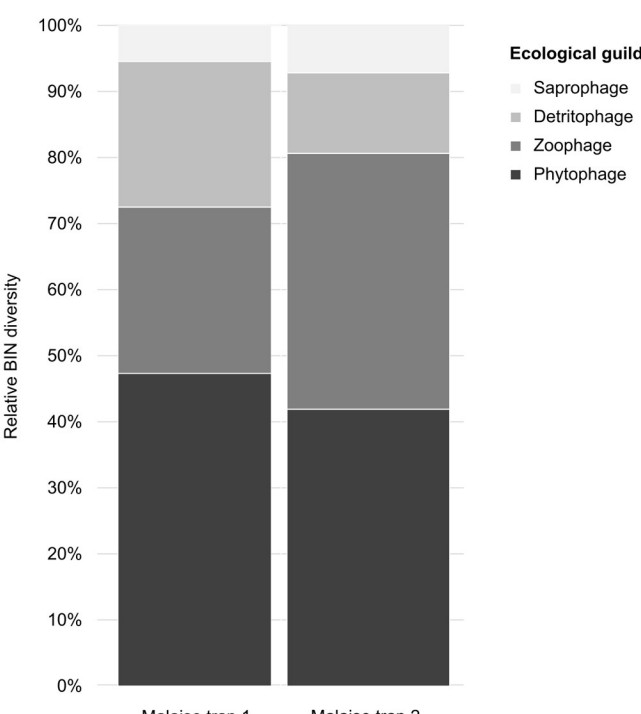

**Fig 4. Relative BIN diversity.** a. Relative BIN diversity across the top most abundant families in our Malaise traps. b. Relative BIN diversity across ecological guilds.

## COI clusters across methods

In total, 77,497 specimens of insects were processed, 52,362 from Malaise trap 1 and 25,135 from Malaise trap 2. Excluding all flagged sequences from analysis (and retaining only those with at least 500 bp) reduced our numbers to 39,374 and 19,394 COI-sequences for each trap respectively. For comparative analysis of cluster algorithms (in terms of cluster diversity), we reran the RESL-algorithm on these sequences which led to the recovery of 6,283 (MT1) and 5,253 (MT2) OTUs that are unique to our project (Table 2). SpeciesIdentifier (using the 3% threshold) suggested slightly fewer clusters than the RESL-algorithm, while ASAP (1st partition) calculated more conservative values, i.e., a much lower number of putative species (Table 2 and Fig 5).

## Discussion

### Overwhelming species richness despite drastic undersampling

All accumulation curves (Figs 3A and 5) and diversity profiles (Fig 3B and 3C) indicate that we have drastically undersampled both trap sites. This was expected for several reasons. First, our collection effort was limited in space and time, using two Malaise traps for three months only. Unlike temperate regions, generally speaking, no individual season in the tropics is highly unsuitable in terms of activity for all arthropod species [41], meaning that arthropods are present and mobile all year round [41, 42]. Therefore, sampling only three months provides a limited coverage of temporal species diversity. Second, while Malaise traps are very effective at collecting arthropods [26], we did not use any additional sampling method to incorporate the diverse canopy communities present in many tropical forests [29, 43, 44]. Our sampling

## OTU diversity per clustering algorithm

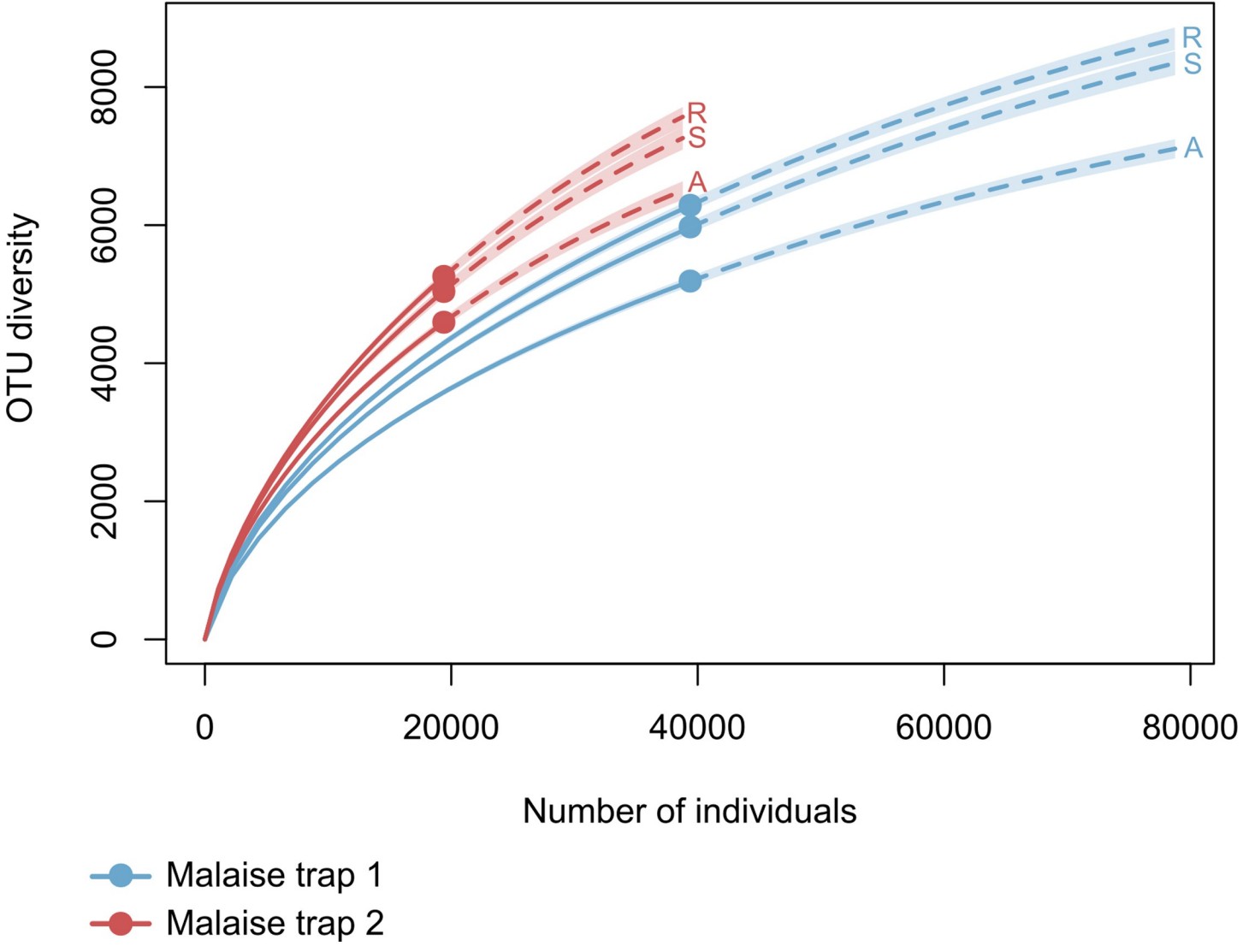

**Fig 5. Accumulation curves of OTU diversity.** Recovered with each clustering algorithm (R: RESL, A: ASAP, S: SpeciesIdentifier) for each Malaise trap. Dotted lines represent extrapolated values (up to double the sampling effort), bold lines represent interpolated values. Shaded areas represent the 95% confidence intervals.

techniques targeted arthropods that are found in the litter and understory habitats, whereas [43] and colleagues have demonstrated that the highest species richness is found in the forest canopy. Third, we did not process all collected individuals due to economic constraints. We had a total of eleven collection events per trap. Seven bulk samples that were collected with Malaise trap 1 were processed entirely; however, sequencing of all other samples was limited to 15 (1,475 specimens) plates per sample. Had we doubled our sampling effort, we would have recovered at least 38% and 44% more putative species for Malaise trap 1 and 2, respectively (Fig 3A). Sampling was slightly more comprehensive with Malaise trap 1, which is presumably due to the fact that more individuals were processed from these samples. Nevertheless, we

clearly only recovered a fraction of the actual diversity present at the sites: Chao1 calculations estimated much higher species numbers for each trap site, and we see no overlap between empirical and estimated BIN numbers for all three diversity orders (species richness q = 0; Shannon diversity q = 1; Simpson diversity q = 2).

## Patchiness in arthropod diversity

Beta diversity assessments show that the communities from each trap site are significantly distinct and that this difference is driven by location effects only (all samples were dispersed homogeneously) (S1 Table and S1 Fig). Even after pooling all collection events together, we observed only 24% overlap in putative species between traps despite the close proximity (< 400 m). One can argue that due to the limited sequencing of sample contents, we are unknowingly comparing two very different subsets of actual similar communities, which was also mentioned by [45] as a factor contributing to the overestimation of beta-diversity [45]. However, we suggest that this is not the case because we recovered more than 80% sample coverage for each Malaise trap. Instead, we argue that we here witness arthropod diversity patchiness, as described by [46]. Forest floors are highly heterogeneous in the tropics over small spatial scales, resulting in high microhabitat richness in x-dimensions [46–48]. Nutrient availability, habitat heterogeneity, spatial variation of plant communities, degree of exposure to predators, and ecosystem disturbances are just some factors that define these microhabitats and their arthropod communities [47, 49, 50].

Prior studies on tropical rainforests have demonstrated that because the majority of insects are herbivores and host-specific, vegetation has a high impact on the prevailing arthropod species, which can account for up to 60% of insect variation [45, 51]. In our study, almost half of all recovered species per trap were phytophages (Fig 4B), meaning that differing vegetation at each trap site could be a driving factor behind the high species turnover [52]. Moreover, because Malaise traps capture insects that happen to fly through a very limited area, various factors such as trap location, orientation, height based on vegetation, light exposure and surrounding structures also have a direct effect on captured communities [53, 54]. In a recent study [54], examined the effects of Malaise trap spacing on species richness and composition, and found that community-similarity decreased among all major taxa with increasing trap-to-trap distances. Also, they found that 18 m between traps was the cut-off value where the number of shared species dropped significantly [54]. These results reinforce our assumption that we are in fact sampling and comparing two very different insect communities with one another.

## Guild structures

Despite recovering a high species turnover between trap sites, community compositions at the family level were very similar (Fig 4A). In consequence, guild structure was also conserved (Fig 4B). However, we highly encourage further research to look into this because we analyzed guild structures only at the family-level. Although it is convenient to place entire families into guilds, it is also a source of error because species of the same families can cover a wide range of feeding behaviors [48]. However, assigning single species to guilds is a major challenge, especially in large-scale surveys. There is too little literature on the feeding activities of single species, and even then, different life stages of the same species can fall into different guild categories (e.g., parasitoid Hymenoptera), and for some taxa, feeding activities of adult species are completely unknown [48]. Also, only a small proportion of our sequences provided identification at the species level, meaning that we cannot apply feeding traits to species proxies. In this study, we did not conduct morphological identifications. Instead, all family-level

identifications were assigned automatically using the identification tool on BOLD. It is therefore important to note that accurate results are only guaranteed provided that high quality reference libraries are being used as a backbone, which include sequences of vouchers that have been accurately identified morphologically. Despite these sources of bias, we still believe that we can rely on these assigned identifications as we are only using them at the family-level, for which extensive information is available on BOLD.

For the family-level guild assignment, we used the Insect Trait Tool that was developed by [37]. Because this tool was developed for the Central European fauna, the extended trait information provided by the tool may not be accurate for tropical fauna. However, because we conducted only a broad guild analysis, we do not think that this is problematic in our study.

## Dark taxa: Abundant, diverse, unknown

In our study, the majority of all BINs were rare, being represented by one or two specimens only (Fig 2). Although we did expect to capture a high proportion of singleton species, we recovered a surprisingly higher frequency of rare species than expected for large-scale tropical surveys, which is typically at about 32% [55, 56]. A closer look at the data revealed that the majority of these singletons are associated with "dark taxa", highly diverse groups of arthropods (mostly Diptera and Hymenoptera) for which little taxonomic or life-history information is available [6, 8]. In total, 70% (40,807) of all processed specimens and 58% (5,340 BINs) of all recovered putative species in this study are shared by eight dark taxa families only, namely Cecidomyiidae (gall midges), Ceratopogonidae (biting midges), Chironomidae (non-biting midges), Phoridae (scuttle flies), Psychodidae (sand flies), Sciaridae (dark-winged fungus gnats), Platygastridae and Braconidae (both parasitoid wasps).

As demonstrated in this study, dark taxa can be highly abundant and often make up the bulk of an insect sample not only in the tropics, but also in temperate regions [6, 57]. With this being a global phenomenon, the inability to associate these insects to species names or ecological functions is a large constraint to biodiversity research, conservation priority setting as well as understanding ecosystem functioning. One recent publication [20] highlighted that dark taxa are so abundant that they should be included in any holistic biodiversity assessment, but tackling them with traditional taxonomic techniques is too slow [20, 58]. Specimens of dark taxa are often small-bodied and cryptic diverse, so often (especially for Diptera), specimens need to be dissected and studied microscopically. Moreover, species identifications for these insects is often only possible with the use of multiple approaches in parallel to ensure accurate results. Integrative approaches that combine various methodologies are therefore becoming ever more important in making these groups tangible to science [4, 20, 59].

Since 2020, the third phase of the nationwide German Barcode of Life project (GBOL III: Dark Taxa; https://bolgermany.de/home/gbol3/de/projekte/) is dedicated to tackling difficult groups of taxa and training a new generation of taxonomists. In this initiative, integrative methods are being used in order to speed up the identification of dark taxa and making them more tangible to science. to do this, researchers are using (among others) a reverse and integrative taxonomical approach to effectively target and study their groups of interest. This consists of first applying molecular methods (including MinION technologies) to rapidly distinguish sequences clusters among thousands of preselected specimens, then applying morphological methods to target specimens of specific clusters for species identification. This technique drastically reduces the workload because time-consuming specimen processing and morphological analysis is drastically reduced. However, this approach it still time consuming, because it still requires the processing of thousands of individuals, as in our case [60]. One technology that is currently expediting biomonitoring surveys is metabarcoding, which allows

the analysis of entire bulk samples in one sequencing run [14, 17]. However, this method only provides information on community compositions and not on abundance data, nor it the link between sequence and specimen conserved [60, 61]. This makes it especially difficult to study dark taxa because they consist of many species that are not yet described, so these remain undescribed because specimens cannot be easily pinpointed [60].

Just recently, new technological developments have emerged which can help accelerate bio-monitoring studies by speeding up the greatest bottleneck of ecological research–sample sorting. Bulk samples of arthropods often contain hundreds to thousands of specimens, that need to be sorted before conducting species-level analyses. In their study, [60] present a compact insect sorting robot which has the ability to recognize and sort insect specimens based on overview images of bulk samples. Especially interesting is the fact this robot, the DiversityScanner, is able to process very small specimens (<3 mm) [60]. Specimens are automatically selected by the scanner, imaged, assigned to a class or family, then moved to a microplate. Another study, [62], propose a workflow that combines HotSHOT with MinION technologies to conduct fast and accurate species-level sorting of ecological samples [62]. With a modest amount of equipment, manpower, and training, the authors were able to conduct species-level sorting within hours, which came down to 2.5 minutes per specimen. Of course, species identification can only be provided if identified sequences are present in databases, however, coupling this approach with the aforementioned reverse workflow that is applied in the GBOL III project could drastically expedite the work for taxonomists. Because no taxonomic expertise is necessary for the laboratory produces, taxonomists can be first brought on board to analyze vouchers after cluster analysis.

## Employing DNA-based delimitation methods: Working with species proxies

BOLD not only provides a variety of analytical- and visualization techniques, its interface is also very user-friendly, making it easy for all researchers (even with little or no bioinformatic knowledge) to use [32]. Due to this, BOLD is commonly used in DNA barcoding research, so consequently, its integrated RESL algorithm and BIN system is also commonly used for sequence data clustering. For our analyses, we used BIN-counts as a proxy for species diversity, as has been done in various studies [6, 23, 63–66]. However, there are varying opinions regarding using BINs for species delimitation [39, 46], especially when assuming that BIN numbers are equal to species numbers in a 1:1 ratio. Therefore, as [67] recommended, we analyzed our sequence data with several species delimitation methods that apply different algorithms to compare the number of clusters recovered with each method. We used SpeciesIdentifier for objective clustering using a preset threshold (3%) for comparative purposes and to increase confidence regarding the relative extent of diversity in our traps. We recovered slightly fewer clusters than with the RESL-algorithm from BOLD [Malaise Trap 1: 5,967 versus 6,283 OTUs; Malaise Trap 2: 5,054 versus 5,253 OTUs]. With ASAP, hierarchical clustering was done using pairwise genetic distances of sequences. The program builds numerous partitions ranked by scores, with the best ones provided in the output to be used for analysis. With ASAP, we obtained much more conservative cluster counts than with the RESL (and SpeciesIdentifier) algorithm, especially for Malaise trap 1 (Table 2 and Fig 5). Analysis across methods displayed similar trends in regard to sample coverage, depicting that the sample contents of Malaise trap 1 were much better sampled than of Malaise trap 2 (Table 2).

## Conclusion

Here, processing only a fraction of bulk samples collected during merely three months of Malaise trap sampling recovered more than 9,000 putative species and high species turnover

among two very close sites. Despite processing more than 77,000 specimens, community analysis suggests that we strongly undersampled both collection sites. Community compositions at the family level were conserved between traps, revealing similar ecological guild functions. The majority of specimens collected and processed belong to the so-called dark taxa, for which little taxonomic and life history information is available. Comprehensive specimen sampling, KI-powered sample processing, and highest throughput sequencing coupled with trait analysis will be crucial to address this knowledge gap, for which the technological is being created now [38, 68–70].

## Supporting information

**S1 Fig. Non-metric dimensional scaling (NMDS) of the community compositions.** NMDS plot of the insect community compositions within each collection sample. Ellipses are 95% confidence intervals of centroids for each Malaise trap.
(PDF)

**S1 Table. Statistical analysis of the community compositions.**
(PDF)

## Acknowledgments

We would like to thank the Ontario Genomics Institute conducting sequence analysis and informatics.

## Author Contributions

**Conceptualization:** Stefan Schmidt, Bruno Cancian de Araujo, Thomas von Rintelen, Olga Schmidt, Hasmiandy Hamid, Raden Pramesa Narakusumo, Michael Balke.

**Data curation:** Bruno Cancian de Araujo.

**Formal analysis:** Caroline Chimeno, Olga Schmidt.

**Funding acquisition:** Michael Balke.

**Investigation:** Stefan Schmidt, Bruno Cancian de Araujo, Thomas von Rintelen, Olga Schmidt.

**Methodology:** Stefan Schmidt, Kate Perez, Thomas von Rintelen, Olga Schmidt, Michael Balke.

**Project administration:** Michael Balke.

**Software:** Caroline Chimeno.

**Visualization:** Caroline Chimeno.

**Writing – original draft:** Caroline Chimeno.

**Writing – review & editing:** Caroline Chimeno, Stefan Schmidt, Hasmiandy Hamid, Raden Pramesa Narakusumo, Michael Balke.

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
