## [Decision Letter · Decision Letter 0]

5 Jun 2023

PONE-D-23-12738Abundant, diverse, unknown: Extreme species richness and turnover despite drastic undersampling in two closely placed tropical Malaise trapsPLOS ONE

Dear Dr. Chimeno,

Thank you for submitting your manuscript to PLOS ONE. After careful consideration, we feel that it has merit but does not fully meet PLOS ONE’s publication criteria as it currently stands. Therefore, we invite you to submit a revised version of the manuscript that addresses the points raised during the review process.

The two invited reviewers have submitted their reports and both are positive about your work. However, they have both identified a number of issues that need to be addressed, and the second reviewer has also raised concerns about the structure and content of the Materials and Methods and Discussion sections, which I think you should consider carefully. 

We look forward to receiving your revised manuscript.

Kind regards,

Pierfilippo Cerretti, Ph.D.

Academic Editor

PLOS ONE

Reviewers' comments:

Reviewer's Responses to Questions

**Comments to the Author**

1. Is the manuscript technically sound, and do the data support the conclusions?

Reviewer #1: Yes

Reviewer #2: Partly

2. Has the statistical analysis been performed appropriately and rigorously? 

Reviewer #1: Yes

Reviewer #2: Yes

3. Have the authors made all data underlying the findings in their manuscript fully available?

Reviewer #1: No

Reviewer #2: Yes

4. Is the manuscript presented in an intelligible fashion and written in standard English?

Reviewer #1: Yes

Reviewer #2: Yes

5. Review Comments to the Author

Reviewer #1: Review of ‘Abundant, diverse, unknown: Extreme 1 species richness and turnover despite

2 drastic undersampling in two closely placed tropical Malaise traps’

I can find very little to criticise in this manuscript. I suppose that none of the results are particularly surprising, as the authors acknowledge, but the empirical demonstration of massive local heterogeneity and very high species richness are very useful, and the methods will be very useful for the development of further studies at different scales. I would hope that the availability of large numbers of barcodes for ‘dark taxa’ would also promote further taxonomic work.

For those samples for which you filled 15 plates, rather than all specimens, what was the protocol for choosing individuals to sequence? Or were specimens chosen randomly until the plates were full?

What happens to the specimens? Are they available for taxonomic work?

I couldn’t find the datasets which are deposited on BOLD. Are they publicly available yet? I was hoping to scan through the areas I know best (Ichneumonoidea) and sense check the results.

Lines 46-47: I don’t think ‘dark taxa’ need be defined by a complete absence of taxonomic expertise, but clearly the expertise is very limited and completely inadequate to the job.

Change ‘an increase by…’, to ‘an increase of…’ (eg., line 205)

Reviewer #2: The manuscript by Chimeno et al. compares the insect communities collected using two closely spaced Malaise traps in Sumatra. It presents a well sampled dataset of ~70K insects and compares these two sites in terms of alpha and beta diversity. The findings of high species diversity of insects along with the distinct communities in nearby sites are of interest for wider scientific community. However, I would suggest the following be considered:

The paper shows surprisingly distinct communities across the two sites despite their proximity. But the manuscript has a very limited description of the sites being compared (other than them being forest edge sites). It would be important to describe the two sites in the study in detail and the area around in general in the methods section of the manuscript.

The manuscript furthermore characterizes the insect communities at family level but does not describe how the family level identifications were made for the insects. This should be properly documented in the methods. Similarly, standard barcode protocols of CCDB is cited at L93 but the link directs to several other links: It would be good if the major steps are described (target region, extractions, sequencing methods) given the linked protocols may change in future. Regarding data analysis, how were the sequences aligned?

While those familiar with the topic would not find the assumption that the specimens sequenced in this project (given they are from Southeast Asia and given the insect families they belong to) are from dark taxa ‘for which little taxonomic… information is available’ (L277) surprising, it is worth noting that the manuscript does not document this finding. It would be valuable to characterize how many of the OTUs sequenced in this project are (1) present in the barcode databases and (2) are identified to species level. This further links to the title of the paper abundant diversity and “unknown”.

A key issue relates to the discussion of the results. The authors find high diversity, distinct communities and that most species and specimens belong to a few families. However the discussion of the solution to these challenges is not developed; i.e. what are the recent methods that can be used to expedite the work for taxonomists and how can such large scale studies be conducted given that even 70K specimens are not sufficient for studying these sites? What is the proposal for future research, particularly in tropics?

This may be a preference, but I find that structurally the first section of the discussion “Employing DNA-based delimitation methods: working with 225 species proxies” is less interesting than several of the key findings described in the paper and it would be more interesting to highlight the key findings of the manuscript first.

Minor:

L152: The links are not working for me

L244: -21% and -14% is confusing to me. 21% of 6283 (RESL) is 1319 and -21% would imply 4960 ASAP OTUs based on table 2, but doesn’t match 5185. Perhaps it should be +21% if the reference is ASAP OTUs. But overall, I would suggest not using % here given the clusters themselves are not necessarily the same. Same for L239.

L278-282: mentions eight families but lists only seven

6. PLOS authors have the option to publish the peer review history of their article (what does this mean?). If published, this will include your full peer review and any attached files.

Reviewer #1: **Yes: **Gavin Broad

Reviewer #2: No

---

## [Author Response · Author response to Decision Letter 0]

19 Jun 2023

Reviewer #1: Review of ‘Abundant, diverse, unknown: Extreme 1 species richness and turnover despite drastic undersampling in two closely placed tropical Malaise traps’

I can find very little to criticise in this manuscript. I suppose that none of the results are particularly surprising, as the authors acknowledge, but the empirical demonstration of massive local heterogeneity and very high species richness are very useful, and the methods will be very useful for the development of further studies at different scales. I would hope that the availability of large numbers of barcodes for ‘dark taxa’ would also promote further taxonomic work.

For those samples for which you filled 15 plates, rather than all specimens, what was the protocol for choosing individuals to sequence? Or were specimens chosen randomly until the plates were full?

Author response: 

Since the main aim was to maximize the number of BINs from the Indonesian Malaise traps, a selective strategy was used to narrow down the number of specimens especially for the samples with only 15 plates processed. Individuals chosen for sequencing were selected to capture as much diversity as possible based on size and morphospecies. Two sizes of sieves were used to subsample from three different size classes (no sieve, 8mm sieve, and 2mm sieve). As most of the diversity was likely hidden in the smaller organisms (particularly the abundant insect orders: Hymenoptera, Coleoptera, and Diptera), more specimens were chosen from the smallest size class. Additionally, more Hymenoptera and Coleoptera were selected as opposed to Diptera because Diptera are often so abundant in Malaise trap samples that there is a higher risk of oversampling the same species. 

We have added this information to the manuscript, see lines 96-105.

What happens to the specimens? Are they available for taxonomic work?

Author response: 

All barcoded specimens are currently stored at the Center for Biodiversity Genomics (CBG) natural history archive (collection code BIOUG) at the University of Guelph, Canada. However, this collection, as well as the rest of the unprocessed material, will eventually be repatriated to Museum Zoological Bogoriense in Cibinong, Indonesia. We have also added this to the manuscript, see lines 114-117.

I couldn’t find the datasets which are deposited on BOLD. Are they publicly available yet? I was hoping to scan through the areas I know best (Ichneumonoidea) and sense check the results.

Author response: 

In the meantime, the DOIs went public and are now accessible under the following links: doi.org/10.5883/DS-GMTINDO1 and doi.org/10.5883/DS-GMTINDO2.

Lines 46-47: I don’t think ‘dark taxa’ need be defined by a complete absence of taxonomic expertise, but clearly the expertise is very limited and completely inadequate to the job.

Author response: 

Thank you, we have made changes. See lines 46-47: …which is either in decline or very limited, the latter being the case in the so-called dark taxa (6,7).

Change ‘an increase by…’, to ‘an increase of…’ (eg., line 205)

Author response: 

Thank you, we have changed this, also throughout the manuscript.

Reviewer #2: The manuscript by Chimeno et al. compares the insect communities collected using two closely spaced Malaise traps in Sumatra. It presents a well sampled dataset of ~70K insects and compares these two sites in terms of alpha and beta diversity. The findings of high species diversity of insects along with the distinct communities in nearby sites are of interest for wider scientific community. However, I would suggest the following be considered:

The paper shows surprisingly distinct communities across the two sites despite their proximity. But the manuscript has a very limited description of the sites being compared (other than them being forest edge sites). It would be important to describe the two sites in the study in detail and the area around in general in the methods section of the manuscript.

Author response: 

We have added more site information, see lines 79-88.

The manuscript furthermore characterizes the insect communities at family level but does not describe how the family level identifications were made for the insects. This should be properly documented in the methods. Similarly, standard barcode protocols of CCDB is cited at L93 but the link directs to several other links: It would be good if the major steps are described (target region, extractions, sequencing methods) given the linked protocols may change in future. Regarding data analysis, how were the sequences aligned?

Author response: 

Thank you for addressing this. We have added the information to the Methods (see data analysis). All family-level identifications were conducted using the BIN taxonomy match tool on BOLD, and all sequences without information were excluded from these analyses. The BIN taxonomy match tool applies taxonomic identifications up to the lowest level of concordant taxonomy within all members of a BIN using the entire BOLD database as a reference library.

We have added more information to the laboratory methods. To summarize, the standard 658 base pair region of the cytochrome c oxidase subunit I (COI) gene was the target region. Steps included tissue lysis, DNA extraction, microplate consolidation into 384-well format (to reduce reagents and costs) before PCR amplification, cycle sequencing, and subsequent Sanger sequence analysis. 

For your interest, more detailed sequencing steps are accessible from each sequence page on BOLD if you click on the LIMS icon at the bottom (see images). 

While those familiar with the topic would not find the assumption that the specimens sequenced in this project (given they are from Southeast Asia and given the insect families they belong to) are from dark taxa ‘for which little taxonomic… information is available’ (L277) surprising, it is worth noting that the manuscript does not document this finding. It would be valuable to characterize how many of the OTUs sequenced in this project are (1) present in the barcode databases and (2) are identified to species level. This further links to the title of the paper abundant diversity and “unknown”.

Author response: 

This is a very interesting point, thank you! We have added information to the manuscript, see lines 183 – 189. More than two-thirds (6,125) of all BINs were unique to BOLD, meaning that they were added for the first time with the upload of these sequences. Of the 58,769 specimens that were successfully sequenced, 961 obtained a species-level identification, providing coverage for 231 species. 

We have also added (now) Figure 2 to the manuscript, depicting the frequency of rare and common BINs within the data, and also the proportion of these “unknown” groups. 

A key issue relates to the discussion of the results. The authors find high diversity, distinct communities and that most species and specimens belong to a few families. However the discussion of the solution to these challenges is not developed; i.e. what are the recent methods that can be used to expedite the work for taxonomists and how can such large scale studies be conducted given that even 70K specimens are not sufficient for studying these sites? What is the proposal for future research, particularly in tropics?

Author response: 

We have added more on this topic, and discuss several possibilities for expediting taxonomic work. See lines 370 -407. 

This may be a preference, but I find that structurally the first section of the discussion “Employing DNA-based delimitation methods: working with 225 species proxies” is less interesting than several of the key findings described in the paper and it would be more interesting to highlight the key findings of the manuscript first.

Author response: 

Thank you, we have rearranged all sections of the manuscript so that we first describe our key findings, then finish off with the comparison of the different clustering analyses.

Minor:

L152: The links are not working for me

Author response: 

In the meantime, the DOIs went public and are now accessible under the following links: doi.org/10.5883/DS-GMTINDO1 and doi.org/10.5883/DS-GMTINDO2.

L244: -21% and -14% is confusing to me. 21% of 6283 (RESL) is 1319 and -21% would imply 4960 ASAP OTUs based on table 2, but doesn’t match 5185. Perhaps it should be +21% if the reference is ASAP OTUs. But overall, I would suggest not using % here given the clusters themselves are not necessarily the same. Same for L239.

Author response: This is a good point. We went ahead and removed the %.

L278-282: mentions eight families but lists only seven

Author response: Thanks!

---

## [Decision Letter · Decision Letter 1]

21 Jul 2023

PONE-D-23-12738R1Abundant, diverse, unknown: Extreme species richness and turnover despite drastic undersampling in two closely placed tropical Malaise trapsPLOS ONE

Dear Dr. Chimeno,

Thank you for submitting your manuscript to PLOS ONE. After careful consideration, we feel that it has merit but does not fully meet PLOS ONE’s publication criteria as it currently stands. Therefore, we invite you to submit a revised version of the manuscript that addresses the points raised during the review process.

We look forward to receiving your revised manuscript.

Kind regards,

Pierfilippo Cerretti, Ph.D.

Academic Editor

PLOS ONE

Journal Requirements:

Reviewers' comments:

Reviewer's Responses to Questions

**Comments to the Author**

1. If the authors have adequately addressed your comments raised in a previous round of review and you feel that this manuscript is now acceptable for publication, you may indicate that here to bypass the “Comments to the Author” section, enter your conflict of interest statement in the “Confidential to Editor” section, and submit your "Accept" recommendation.

Reviewer #2: (No Response)

2. Is the manuscript technically sound, and do the data support the conclusions?

Reviewer #2: Yes

3. Has the statistical analysis been performed appropriately and rigorously? 

Reviewer #2: Yes

4. Have the authors made all data underlying the findings in their manuscript fully available?

Reviewer #2: Yes

5. Is the manuscript presented in an intelligible fashion and written in standard English?

Reviewer #2: Yes

6. Review Comments to the Author

Reviewer #2: The authors have addressed my comments. Upon review, and looking through the links that are now available I have a few suggestions.

The numbers in BOLD datasets doi.org/10.5883/DS-GMTINDO1 and doi.org/10.5883/DS-GMTINDO2 correspond to 52239 and 24938 specimens (and strangely enough, sequences) which is slightly different from the numbers in L254 (52362 and 25135) and the actual number of sequences is far fewer. The downloaded sheets from figshare correspond to the values in the manuscript. Based on "BIN_data" from figshare, approximately 12,000 specimens have 0bp barcodes, but the number of sequences in the BOLD page seems to be too high, though upon download the sequence count gets closer, though specimen count is still off. I am not sure what is going on, it would be good if a field of sequence is added to one of the figshare spreadsheets for straightforward access to data for future users along with a flag of which sequences were used for final analysis.

The authors have stated that BOLD taxonomy is used for family level classification. Given these were not morphologically examined/verified, it would be good to mention in the discussion this limitation/potential errors propogating in sequence based identifications.

L361: Edit to "in temperate regions", and given temperate samples are not involved in this study, this should be cited, and the similarity in pattern has been quantified in https://doi.org/10.1038/s41559-023-02066-0

7. PLOS authors have the option to publish the peer review history of their article (what does this mean?). If published, this will include your full peer review and any attached files.

Reviewer #2: No

---

## [Author Response · Author response to Decision Letter 1]

31 Jul 2023

Reviewer #2: The authors have addressed my comments. Upon review, and looking through the links that are now available I have a few suggestions.

The numbers in BOLD datasets doi.org/10.5883/DS-GMTINDO1 and doi.org/10.5883/DS-GMTINDO2 correspond to 52239 and 24938 specimens (and strangely enough, sequences) which is slightly different from the numbers in L254 (52362 and 25135) and the actual number of sequences is far fewer. The downloaded sheets from figshare correspond to the values in the manuscript. Based on "BIN_data" from figshare, approximately 12,000 specimens have 0bp barcodes, but the number of sequences in the BOLD page seems to be too high, though upon download the sequence count gets closer, though specimen count is still off. I am not sure what is going on, it would be good if a field of sequence is added to one of the figshare spreadsheets for straightforward access to data for future users along with a flag of which sequences were used for final analysis.

Response: This is in fact a simple mistake – we realized that the BOLD support team has not transferred all specimens to the dataset that was published. We have contacted them and this will be changed!

The authors have stated that BOLD taxonomy is used for family level classification. Given these were not morphologically examined/verified, it would be good to mention in the discussion this limitation/potential errors propogating in sequence based identifications.

Response: This has been added to the discussion.

L361: Edit to "in temperate regions", and given temperate samples are not involved in this study, this should be cited, and the similarity in pattern has been quantified in https://doi.org/10.1038/s41559-023-02066-0

Response: Thank you, we have added this information also.

---

## [Editor Report · Decision Letter 2]

3 Aug 2023

Abundant, diverse, unknown: Extreme species richness and turnover despite drastic undersampling in two closely placed tropical Malaise traps

PONE-D-23-12738R2

Dear Dr. Chimeno,

We’re pleased to inform you that your manuscript has been judged scientifically suitable for publication and will be formally accepted for publication once it meets all outstanding technical requirements.

Kind regards,

Pierfilippo Cerretti, Ph.D.

Academic Editor

PLOS ONE
---

## [Editor Report · Acceptance letter]

7 Aug 2023

PONE-D-23-12738R2 

Abundant, diverse, unknown: Extreme species richness and turnover despite drastic undersampling in two closely placed tropical Malaise traps 

Dear Dr. Chimeno:

I'm pleased to inform you that your manuscript has been deemed suitable for publication in PLOS ONE. Congratulations! Your manuscript is now with our production department. 

Kind regards, 

on behalf of

Dr. Pierfilippo Cerretti 

Academic Editor

PLOS ONE